# Calretinin Functions in Malignant Mesothelioma Cells Cannot Be Replaced by the Closely Related Ca^2+^-Binding Proteins Calbindin-D28k and Parvalbumin

**DOI:** 10.3390/ijms19124015

**Published:** 2018-12-12

**Authors:** Janine Wörthmüller, Anne Oberson, Valérie Salicio, Walter Blum, Beat Schwaller

**Affiliations:** 1Anatomy, Section of Medicine, University of Fribourg, Route Albert-Gockel 1, 1700 Fribourg, Switzerland; janine.woerthmueller@unifr.ch (J.W.); anne.oberson@unifr.ch (A.O.); valerie.salicio@unifr.ch (V.S.); blum@genetica-ag.ch (W.B.); 2Genetica AG, 8001 Zurich, Switzerland

**Keywords:** malignant mesothelioma, calretinin, calbindin, parvalbumin

## Abstract

Calretinin (CR; *CALB2*) belonging to the family of EF-hand Ca^2+^-binding proteins (CaBP) is widely used as a positive marker for the identification of human malignant mesothelioma (MM) and functionally was suggested to play a critical role during carcinogenesis of this highly aggressive asbestos-associated neoplasm. Increasing evidence suggests that CR not only acts as a prototypical Ca^2+^ buffer protein, i.e., limiting the amplitude of Ca^2+^ signals but also as a Ca^2+^ sensor. No studies have yet investigated whether other closely related CaBPs might serve as substitutes for CR’s functions(s) in MM cells. Genetically modified MM cell lines with medium (MSTO-211H and ZL5) or low (SPC111) endogenous CR expression levels were generated that overexpress either CR’s closest homologue calbindin-D28k (CB) or parvalbumin (PV), the latter considered as a “pure” Ca^2+^ buffer protein. After lentiviral sh*CALB2*-mediated CR downregulation, in both MSTO-211H and ZL5 cells expressing CB or PV, the CR deficiency-mediated increase in cell death was not prevented by CB or PV. With respect to proliferation and cell morphology of SPC111 cells, CB was able to substitute for CR, but not for CR’s other functions to promote cell migration or invasion. In conclusion, CR has a likely unique role in MM that cannot be substituted by “similar” CaBPs.

## 1. Introduction

Calcium ions (Ca^2+^) are one of the most important second messengers in eukaryotic cells and are mediators of a great variety of cellular functions. Cells are equipped with sophisticated machinery to precisely regulate the shape (amplitude, duration) of Ca^2+^ signals in a localization-specific manner, e.g., by the action of a multitude of Ca^2+^-binding proteins (CaBPs) located mostly in the cytoplasm [1]. The EF-hand CaBPs are a family of more than 240 proteins encoded by the human genome [2] that include the proteins parvalbumin (PV), calbindin-D28k (CB), and calretinin (CR). CaBPs differing in Ca^2+^-binding affinity and kinetics are known to exert their specific effects at precise temporal windows of Ca^2+^ signaling and moreover are differentially expressed within subpopulations of neurons [3] but also in non-neuronal cell types. Classically they are classified as Ca^2+^ buffers, acting as transitory Ca^2+^ sinks/stores, or as Ca^2+^ sensors, i.e., intracellular proteins that undergo considerable conformational changes upon Ca^2+^-ion binding resulting in interaction(s) with target molecules [4]. PV (M_r_ ~12 kDa; human gene symbol *PVALB*), is considered a pure and typical “slow” Ca^2+^ buffer [2], while CB (M_r_ ~29 kDa; human gene symbol *CALB1*) binds Ca^2+^ ions with medium/high affinity. Numerous in vitro studies have shown that CB, besides functioning as a fast Ca^2+^ buffer, has additional Ca^2+^ sensor functions. Several target proteins of CB have been identified, including plasma membrane ATPases, caspase-3, inositol-monophosphatase (IMPase) and TRPV5, among others (reviewed in [4,5]).

Although CR (M_r_ ~31 kDa; human gene symbol: *CALB2*) was at first considered a typical Ca^2+^ buffer protein for many years, increasing evidence suggests that CR also acts as a Ca^2+^ sensor protein [6,7,8]. CR’s binding targets include the pore-forming α_1_ subunit of the Ca^2+^ channel Ca_V_2.1 [9], the mutant form of huntingtin (mHtt) in neurons [10] and cytoskeletal elements such as cytokeratins and α-tubulin in WiDr colon cancer cells [11]. In a recent study, CR was shown to co-immunoprecipitate with septin 7, and moreover, to co-localize with this protein during cytokinesis in distinct regions of the cleavage furrow of malignant mesothelioma (MM) mitotic cells, supporting a role of CR in the mitotic process [12]. In addition, in the same cells, CR was found to interact with focal adhesion kinase (FAK) [13], a protein known to promote tumorigenesis and metastasis in several cancers [14]. CR is widely used as a diagnostic marker for the identification of epithelioid and mixed (biphasic) MM [15,16], a highly aggressive cancer that arises from mesothelial cells covering the surfaces of the pleura, peritoneum, and pericardium that is typically related to exposure to mineral fibers such as asbestos [17]. CR is suggested to have a pivotal role in the initiation process of mesotheliomagenesis [18,19], is essential for MM cell growth and survival in vitro [20] and moreover, promotes migration, invasion and epithelial-to-mesenchymal transition (EMT) of MM cells [13]. Downregulation of CR decreases the survival of MM cells in vitro [20] and impairs tumor progression in vivo in a mouse xenograft model based on the intraperitoneal injection of human MM cells [13]. Thus, CR has been proposed as a new putative therapeutic target for MM treatment.

CR and CB share 59% amino acid identity and 77% similarity [21,22] and together with secretagogin form the “calbindin” subfamily, all comprising 6 EF-hands. Due to their distinct mostly non-overlapping expression patterns, e.g., CR in cerebellar granule cells and CB in Purkinje cells, and based on findings obtained from studies in knockout mice, it appears that these two proteins are functionally distinct [23]. Nevertheless, the high similarity of the two proteins at the amino acid level suggests that CR and CB might possibly interact with common/similar targets and/or have some shared biological functions. Although CB is not endogenously expressed in mesothelial cells or MM, to our knowledge no studies have previously addressed whether CB could potentially exert CR’s functions in MM cells. To answer this question, and to elucidate whether the Ca^2+^ buffering capacity of CR, or on the contrary, its Ca^2+^ sensor function is implicated in the proliferation/survival of MM cells, we overexpressed the CaBPs (I) PV, considered as a “pure” Ca^2+^ buffer (II) CB, a Ca^2+^ buffer protein with reported Ca^2+^ sensor functions, and (III) CR in different MM cell lines with various endogenous CR expression levels and performed a set of “rescue experiments” by downregulating CR expression *via* a lentiviral approach.

## 2. Results

### 2.1. Generation and Characterization of MM Cell Clones Overexpressing the CaBPs CR, GFP-CR, CB and PV

Lentiviral transduction of different CaBPs was carried out in three human MM cell lines: MSTO-211H, composed of mostly epithelioid cells and some sarcomatoid (spindle-shaped) cells and with relatively high CR levels; ZL5 with epithelioid morphology and intermediate-to-low CR levels and finally SPC111, also biphasic, however, with a large proportion of spindloid cells and characterized by very low endogenous CR expression levels [20]. From the cell lines ZL5 and SPC111 overexpressing the various CaBPs, a minimum of two to four clones were isolated by limited dilution, and the amount of overexpressed protein was determined semi-quantitatively by Western blot analyses as exemplified for the cell line SPC111 (Figure 1A). Details for ZL5 clones are shown in Appendix A. Cloning by serial dilution was not successful for MSTO-211H cells, yet CR levels in ZL5-CR and SPC111-CR clones were clearly higher than in MSTO-211H wt cells (Appendix A). For each clone, the amount of loaded cytosolic extracts was adjusted to the linear range of the Western blot signals obtained with the pure proteins (1.5–10 ng for CR and CB and 1–3 ng for PV). CaBP concentrations for all clones were calculated from the standard curves and multiplied by the number of functional Ca^2+^-binding sites within a given protein: five for CR, four for CB, and two for PV. We aimed to select groups of clones with the expression of a similar amount of “Ca^2+^-binding sites” in terms of their global Ca^2+^-buffering capacity. The calculated values for the three groups of CaBP-overexpressing clones are shown for SPC111 cells (Figure 1B). In the group of CR clones, the concentration of Ca^2+^-binding sites ranged from 90 to 280 µM (average: 180 µM). Similar, but slightly lower concentrations were observed in CB clones (70–150 µM; average: 102.5 µM). Lower concentrations of Ca^2+^-binding sites were detected in the three PV clones (average: 5 µM), i.e., ≈20–40-fold lower than in the CB and CR clones, respectively. In addition, low PV expression levels in PV-overexpressing clones were also detected in ZL5 PV-clones (Appendix A), possibly indicating that high exogenous levels of PV are not well tolerated in the cell lines tested. Thus, this precluded a direct comparison between clones expressing PV and the other two CaBPs with respect to the effect of the Ca^2+^-buffering capacity. Of note, none of the cell lines used in this study expresses CB or PV endogenously at levels detectable by Western blot analysis, yet strongly overexpressed the two proteins in the respectively selected clones, as demonstrated for clones derived from SPC111 cells (Figure 1C).

In all selected clones, CR was downregulated by infection with an LV producing an shRNA directed against *CALB2* resulting in lower CR expression levels 96 h post-infection as exemplified in MSTO-211H parental (wild-type; wt) cells (Figure 1D), in line with previous studies [20]. Treatment of the same cells with an sh*GFP* LV had no effect on CR protein levels. To prove the functionality of the sh*GFP* RNA, MSTO-211H cells overexpressing GFP-CR infected with a sh*GFP* LV showed a strong decrease in the green fluorescence intensity resulting from GFP-CR downregulation (Figure 1E, lower panel) without affecting endogenous CR levels (as shown previously [20]) and without an effect on cell morphology (Figure 1E, upper panels). Cells remained mostly with an epithelioid morphology and proliferation/cell viability was unaffected (Appendix A). On the contrary GFP-CR MSTO-211H cells treated with a sh*CALB2* LV resulted in a considerable decrease in the number of viable cells (Figure 1E) and in the proliferation rate (Appendix A). The essentially unchanged green fluorescence intensity in the remaining cells indicated that those cells were probably not infected by the LV. Based on the absence of an effect induced by sh*GFP*, in all further experiments, sh*GFP* LV was used as a control for the normalization of the results. In addition, CB and PV levels were evaluated after CR downregulation. Importantly, in MSTO-211H cells overexpressing exogenously either CB or PV, levels of both proteins were unaffected after CR downregulation by sh*CALB2* (Appendix A), confirming the specificity of the used sh*CALB2* RNA for CR.

### 2.2. Calbindin-D28k Rescues the Viability Phenotype of SPC111 Cells Caused by LV-Mediated CR Downregulation

CR downregulation was previously shown to significantly reduce the cell growth and viability of MM cells, particularly in cell lines derived from epithelioid and mixed MM with high CR levels, such as MSTO-211H cells, while the effect of its downregulation is weaker and delayed in cells with endogenously lower levels of CR, such as in SPC111 cells [20]. In line with this study, we observed a decrease of viable cells by ≈50% in the parental (wt) MSTO-211H cells 96 h after sh*CALB2* LV infection (Figure 2A). A similar decrease (≈50%) was observed in parental ZL5 cells of epithelioid origin with visibly lower CR levels than MSTO-211H cells (Figure 2B), and the effect in SPC111 cells was clearly smaller (Figure 2C), both results are in agreement with our previous study [20]. In MSTO-211H cells overexpressing CR, viability after sh*CALB2* treatment was essentially unaffected (Figure 2A). This is attributed to the fact that although CR levels in MSTO-CR cells were considerably down-regulated by the sh*CALB2* RNA (estimated to be >50% (Appendix A)), the remaining CR expression levels were still similar or even higher than the endogenous CR levels in MSTO-211H wt cells (Appendix A), thus being a likely explanation for the unchanged cell viability. In CB-expressing cells the viability was strongly decreased, suggesting that CB may possibly compete with CR with respect to particular functions of CR implicated in cell proliferation/survival (e.g., interaction with FAK and/or septin 7; see below Figure 3). As expected from the low levels of PV overexpression (Figure 1A), cell viability of wt and PV-expressing MSTO-211H cells was not different after CR downregulation (Figure 2A). Concordantly, no “protective” effect of PV expression was also seen in the other two cell lines, ZL5 and SPC111 (Figure 2B,C). While CB had no visible effect on ZL5 cell viability (Figure 2B), all three SPC111 clones expressing CB showed increased viability/survival compared to SPC111 wt cells, significantly seen in clones CB1 and CB2 (Figure 2C). Thus, we hypothesize that CB might affect the same signaling pathways and potentially bind to the same target(s) as CR, either decreasing or increasing cell viability evidenced in MSTO-211H and SPC111 cells, respectively.

Based on the “protective” effect of CB in SPC111-CB clones, we additionally analyzed the morphology and cell growth curves of the different CaBP-expressing clones focusing on SPC111 cells. The morphology of CB-expressing cells was similar to wt cells (Figure 2D), characterized by cuboidal/polygonal morphology and well-defined and compact clusters. CR-overexpressing cells were at times more loosely attached and often showed a more dispersed cell organization (Figure 2D), possibly related to a function of CR in promoting EMT [13]. After sh*CALB2*-mediated CR downregulation for 96 h, the differences observed in wt and CR-expressing cells compared to non-treated cells were more notable. In addition, the morphology of wt and CR-expressing cells were almost identical after the sh*CALB2* treatment, exhibiting more disorganized clusters with few cells undergoing apoptosis, the latter also previously shown for the sarcomatoid MM cell line ZL34 [20] and a higher proportion of spindle-like cells. CB-expressing cells on the contrary, showed a similar morphology as wt (non-treated or sh*GFP*-treated) cells, characterized by the typical cuboidal morphology; however, instead of forming the typical clusters, cells were more dispersed (Figure 2D). While a slight decrease in proliferation was observed between sh*CALB2*-treated compared to sh*GFP*-treated in SPC111 wt cells (Figure 2E, upper panel), no significant differences regarding proliferation were observed in CB-expressing SPC111 cells (Figure 2E, lower panel). SPC111-CR and SPC111-PV cells showed similar growth curves as the SPC111 wt cells, with a decrease in proliferation when infected with sh*CALB2* RNA compared to sh*GFP*-expressing cells (data not shown). This indicates that particularly CB is able to compensate for the consequences caused by a decrease in CR expression levels. However, and unlike CR effects in promoting migration and invasion when overexpressed in SPC111 cells [13], initial experiments demonstrate that CB is ineffective in increasing the migration or invasion of these cells (Appendix A; for details see Supplemental Methods).

### 2.3. Calbindin-D28k Shares Common Targets with Calretinin in MM Cells

CR was previously shown to co-localize with FAK at focal adhesion sites [13], multi-protein structures that link the extracellular matrix to the actin cytoskeleton, essential for cell adhesion, spreading and migration, as well as for cell proliferation and survival [24]. Likewise, SPC111-CB cells showed co-localization of CB with FAK at focal adhesions (Figure 3A). Since co-IP experiments had revealed a physical interaction between CR and FAK [13], we investigated whether CB might co-immunoprecipitate with FAK (Figure 3B). CR was also reported to co-immunoprecipitate with septin 7 [12], a cytoskeletal-related protein implicated in cytokinesis [25]. In line, extracts from CB-expressing SPC111 cells revealed a physical interaction between CB and FAK and also septin 7 (Figure 3B), demonstrating that CB is able to bind to some of the identified targets of CR.

## 3. Discussion

CB and CR are both proteins classified as Ca^2+^ buffer proteins; however, increasing evidence from different studies has demonstrated that both proteins also act as Ca^2+^ sensors. Each protein comprises six E-helix-loop-F-helix-hand (EF)-hand motifs, four being functional in CB and five in CR. However, the domain organization of EF-hand motifs in CR differs from CB [26]. Comparison with the homologous region of CB shows a small difference in sequence identity of a linker that binds the two spatially localized EF-hand loops. This significantly alters the tertiary structure and biochemical characteristics of both CaBPs and may explain the different cellular functions that have been described for both proteins [27]. In addition, the EF-hand 5 of CR has been proposed to be the site where CR interacts with its binding partners [28], while in CB it is the region comprising the second helix (α4) of EF-hand 2 and the linker region to EF-hand domain 3 which are proposed to mediate the interaction [29]. Interestingly, the EF-hand domain 5 of CR is the low-affinity site (K_D,Ca_ ≈ 1.5 mM) unlikely to play a role in Ca^2+^ buffering in physiological conditions, and in CB it is the non-functional EF-hand domain 2. Thus, this allows these two proteins to exert their dual roles as Ca^2+^ buffers and Ca^2+^ sensors simultaneously.

The fact that CR and CB share 59% amino acid identity and 77% similarity [21,22] suggests that they might also share some common functional features. In retinal ganglion cells, a clear compartmentalization of CB and CR distribution has been described, suggesting that the two proteins may perform distinct functions in localized Ca^2+^ signaling [3]. So far, no studies have addressed whether both proteins have similar or overlapping functions in cancer cells, or if CR’s function could potentially be replaced by another CaBP, particularly in MM cells.

Clones expressing PV, a presumably “pure” Ca^2+^ buffer, showed no increased survival after CR downregulation in any of the tested cell lines. Regarding CB, a known Ca^2+^ buffer and sensor protein, similarly to PV, its expression does not increase the survival of MSTO-211H and ZL5 cells in CR-depleted cells. We conclude that neither CB nor PV seem to have the capacity to rescue the effect of CR downregulation in cells with prevalent epithelioid morphology characterized by high-to-intermediate CR expression levels (ZL5, MSTO-211H), pointing towards a likely unique role of CR in those cells and reinforcing the idea that epithelioid MM cell lines are much more dependent on the presence of CR than sarcomatoid MM cells [20]. Overexpression of CB, nevertheless, increased the viability of cells treated with sh*CALB2* in the SPC111 sarcomatoid cell line, implying that CB might be able to replace CR with respect to specific cellular functions in this particular cell line. In addition, CB was shown to share common targets with CR, FAK, and septin 7, and to colocalize with FAK at focal adhesion sites. However, although CB colocalized with FAK at the focal adhesions of SPC111-CB cells, initial experiments demonstrated that CB was unable to exert additional cellular functions of CR, such as promoting the migration or invasion of the cells. The fact that CB and not PV (a “pure” Ca^2+^ buffer) rescued the effect of CR downregulation indicates indirectly that both the Ca^2+^ buffering and the Ca^2+^ sensor functions of CB might be implicated in mediating MM cell survival. However, due to the significantly lower concentrations of Ca^2+^-binding sites that were present in the different PV clones tested (≈20–40-fold lower than the CB and CR clones, Figure 1B), we cannot unambiguously conclude that the increased survival observed in SPC111-CB clones is specifically mediated by the Ca^2+^ sensor function of CB. More experiments are necessary to support such a conclusion.

## 4. Material and Methods

### 4.1. Cell Culture

Human mesothelioma (MSTO-211H), HeLa and HEK293T cells were obtained from the American Type Cell Collection (ATCC, Rockville, MD, USA). The human mesothelioma cell lines SPC111 and ZL5 were obtained from the University Hospital of Zurich (Zurich, Switzerland) [30]. HeLa and HEK293T cells were maintained in DMEM medium supplemented with 10% FBS (Gibco, Basel, Switzerland) and 1% Penicillin/Streptomycin solution (1% PS; Gibco); all others in RPMI-1640 (Sigma-Aldrich, Buchs, Switzerland) containing 10% FBS supplemented with 2.5 μg/mL Amphotericin B (Corning, NY, USA).

### 4.2. Lentiviral (LV) Constructs, Vector Production, and Lentivirus Isolation

For CR overexpression, the previously described plasmid pLV-CALB2 was used [31] and for CR downregulation a plasmid containing the *CALB2* shRNA as described before [20]. The plasmid pLKO.1-sh*GFP* was purchased from Addgene (#30323) (Watertown, MA, USA). For PV overexpression, the pLVTHM-PV lentiviral vector (Addgene plasmid #12247) [32] was used. A lentiviral plasmid coding for a fusion protein consisting of CR N-terminally tagged with the green fluorescent protein (GFP) [20] was used as a control (GFP-CR). The pLVTHM-CB lentiviral vector was used to stably express calbindin-D28k. Briefly, the GFP cassette in pLVTHM (Addgene plasmid #12247) was replaced with rat calbindin-D28k cDNA coding for full-length CB. The cDNA fragment was synthesized by PCR using the primers 5′-AGTCGTTTAACATGGCAGAATCCCACCTGCAG-3′ and 5′-AGTCACTAGTCTAGTTGTCCCCAGCAGAGAG-3′. The amplicon was digested with PmeI and SpeI and inserted into the PmeI and SpeI unique sites of the pLVTHM vector. The lentivirus particles were produced as described before [20]. Briefly, HEK293T cells were co-transfected by the CaPO_4_ method with 3 µg of the envelope plasmid pMD2.G-VSVG (Addgene plasmid #12259), 8 µg of the packaging plasmid psPAX2 (Addgene plasmid #12260), and 10 µg of the transfer plasmid. Lentivirus in the supernatant of HEK293T cells was harvested 48 and 72 h after transfection. The supernatant was filtered (0.45 µm) and resuspended in DMEM containing 10% FBS and 1% PS solution.

### 4.3. LV Titration by Limiting Dilution

sh*CALB2*-containing lentiviral particles at dilutions of 10^−3^ to 10^−7^ were used to infect HeLa cells (50,000 cells/well). The medium was replaced after 48 h with a selection medium containing 2 µg/mL puromycin (Sigma), and at day 12 post-infection, cells were stained with crystal violet to determine the lentiviral titer.

### 4.4. Establishment of Stably Transduced Cell Lines and Western Blot Analysis

pLV-CALB2, pLVTHM-CALB1, pLVTHM-PV, and pLVTHM-eGFP-CR lentiviral vectors were used to stably express CR, CB, PV, and GFP-CR in the human MM cell lines MSTO-211H, ZL5 and SPC111. The expression levels of the different CaBPs were determined by Western blot analysis. Briefly, cells were grown to 70–80% confluence, then trypsinized and protein extracts (cleared lysates) were resuspended in a homogenization buffer containing Tris-HCl (Sigma-Aldrich), 1 mM EDTA (Sigma) and a protease inhibitor cocktail (Quartett GmbH, Berlin, Germany). The suspension was sonicated and centrifuged at 12,000× *g* for 30 min at 4 °C. Protein concentration was determined using the Bradford Assay (Bio-Rad Laboratories AG, Cressier, Switzerland). Proteins (40 µg) were separated by 10% SDS-PAGE, transferred onto nitrocellulose membranes, blocked with 5% BSA in TBS-Tween for 1 h and incubated overnight at 4 °C with the following primary antibodies diluted 1:10,000 in 2% non-fat milk in TBS-T: rabbit anti-Calretinin (CR 7699/4, Swant, Marly, Switzerland), rabbit anti-PV25 (Swant) and rabbit anti-Calbindin-D28k (Swant). Membranes were washed and incubated for 1 h with secondary goat anti-rabbit (HRP)-labeled antibodies (Sigma) at a dilution of 1:10,000. The signals were detected as described in [20].

### 4.5. Clonal Selection

Clonal selection of lentivirus-infected MM cells was performed with the cell lines SPC111 and ZL5. Cells overexpressing the different CaBPs (CR, CB, PV) were counted and diluted to a concentration of 1 cell/100 µL. 100 µL of cell suspension was plated in 96-well plates and individual wells were checked to verify that only one cell was present per well. Up to 5 clones (numbered from 1 to 5) for each CaBP were selected and expanded.

### 4.6. Semi-Quantification of the Different CaBPs-Expressing Clones

Protein samples from MM cell clones expressing the different CaBPs were loaded on the same gel (10–15% SDS-PAGE) together with 1–10 ng of recombinant proteins (CR, CB, and PV), the latter serving to generate standard curves. Signals were normalized using the Ponceau Red S staining intensities, and the amount of CaBPs (ng/mg of protein extract) was accordingly calculated. Final concentrations of EF-hand sites were calculated proportionally according to the number of functional Ca^2+^-binding sites within a given protein; a factor of 2 was used for PV, 4 for CB, and 5 for CR. Groups of clones were selected on the basis of a similar Ca^2+^-buffering capacity to carry out the rescue experiments.

### 4.7. Downregulation of CR in the Different CaBPs-Expressing Clones Using LV-Mediated shRNA In Vitro

The different selected CaBP-expressing clones were seeded in 96-well plates (1000 cells/well) and grown for 24 h. LV containing *CALB2* or *GFP* shRNAs (used as control) was added with a multiplicity of infection (MOI) of 10. 72 h later cells were subjected to an MTT assay to ascertain the number of viable and proliferating cells [33]. In addition, cells were scanned at a 2-h frequency to obtain real-time growth curves and to monitor morphological changes using the Incucyte™ Live-cell Imaging System (Essen Bioscience Inc., Ann Arbor, MI, USA). The reported MTT results are the average from at least 4 independent experiments; each sample was measured in triplicates. Mean and standard deviation are shown in the figures. The statistical significance was calculated using a one-way ANOVA test (GraphPad Software Inc., San Diego, CA, USA). *p*-Values of less than 0.05 were considered significant.

### 4.8. Immunofluorescence

Cells were seeded on 12-mm diameter glass coverslips and when they had reached approximately 60–70% confluence, cells were fixed for 30 min with 4% paraformaldehyde, blocked with TBS-containing donkey serum (10%) and incubated overnight at 4 °C with mouse anti-CB (1:1000, Swant) and rabbit anti-FAK (1:50; Cell Signaling Technology, Danvers, MA, USA) antibodies diluted in Tris-buffered saline (TBS 1X). After washing, cells were incubated for 3 h at room temperature with the following secondary antibodies: Alexa Fluor 488-conjugated donkey anti-rabbit IgG (1:100; Jackson Immunoresearch Laboratories, West Grove, PA, USA) and Cy3-conjugated donkey anti-mouse (IgG) (1:100; Jackson Immunoresearch Laboratories). 4′,6-diamidino-2-phenylindole (DAPI; 5 μg/mL; Molecular Probes, Eugene, OR, USA) was used to stain nuclear DNA and coverslips were mounted with Hydromount solution (National Diagnostics, Atlanta, GA, USA). Images were acquired using a LEICA fluorescent microscope DM6000B (Wetzlar, Germany) equipped with a Hamamatsu camera C4742-95 (Bridgewater, NJ, USA).

### 4.9. Co-Immunoprecipitation

Cells at a confluence of 70% were lysed with 1 mL ice-cold lysis buffer (150 mM NaCl, 1% Triton X-100, 50 mM Tris HCl, pH 8.0) supplemented with protease and phosphatase inhibitors. Following a 30-min incubation period, cells were centrifuged (10,000× *g* for 10 min at 4 °C) and the supernatant collected. Co-IP experiments were performed as described before [13]. In brief, to pull down CB, 2–4 µg of rabbit anti-CB (Swant) was used, then 100 µL of µMACS protein A MicroBeads (Miltenyi Biotec, Auburn, AL, USA) were added to the lysate and incubated at 4 °C for 30 min. Samples were loaded on MACS separation columns (Miltenyi Biotec) and subjected to magnetic immunoprecipitation. The columns were washed 3 times with a wash buffer and protein complexes were eluted in 50 µL of pre-warmed SDS gel loading buffer 1X, subjected to electrophoresis (SDS-PAGE) and subsequent Western blotting. Membranes were probed with the antibodies rabbit anti-CB (1:10,000, Swant), rabbit anti-FAK (1:1000, Cell Signaling Technology), and rabbit anti-septin 7 (Bethyl Laboratories Inc., Montgomery, TX, USA); a mouse anti-paxillin antibody (1:2000; BD Bioscience, Allschwil, Switzerland) served as a negative control.

## Figures and Tables

**Figure 1 ijms-19-04015-f001:**
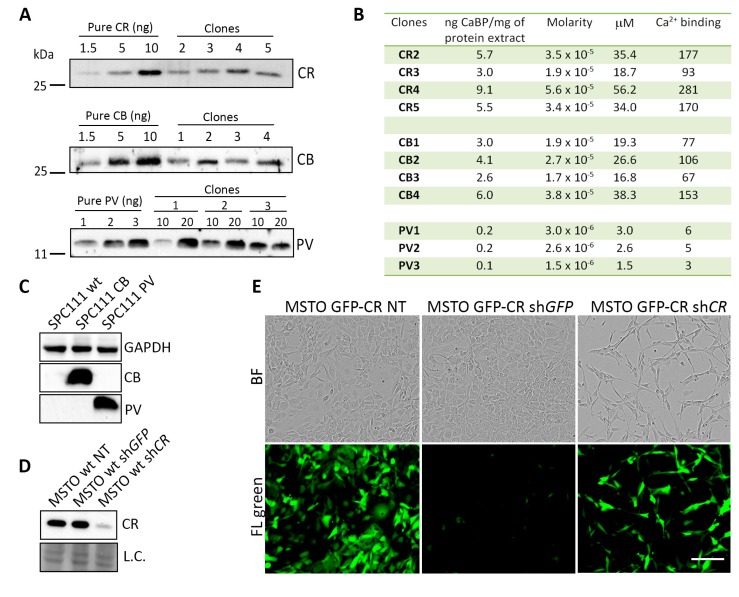
Estimation of the total Ca^2+^-binding capacity provided by the different Ca^2+^-binding proteins (CaBP)-overexpressing clones (exemplified in SPC111 cells) and validation of calretinin (CR) downregulation. (**A**) Protein expression levels of CR, calbindin-D28k (CB), and parvalbumin (PV) in SPC111 clones obtained by serial dilution by Western blot analyses. Semi-quantification was performed using purified recombinant CR, CB, and PV (1 to 10 ng), and calculating a linear regression line; (**B**) Estimated intracellular concentrations in SPC111 CaBP-overexpressing clones. For calculating “Ca^2+^-binding capacity”, concentrations were multiplied by the number of functional EF-hand sites (two for PV, four for CB and five for CR); (**C**) Western blot analysis of SPC111-wt, CB- and PV-overexpressing cells probed simultaneously with CR, CB, and PV antibodies. SPC111-wt cells do not express CB or PV endogenously; (**D**) Western blot analysis demonstrating CR downregulation after 4 days of sh*CALB2* treatment, but not after sh*GFP* transduction in MSTO-211H-wt cells. Ponceau Red staining was used as loading control; (**E**) MSTO-GFP-CR cells treated with sh*GFP*-LV and sh*CALB2*-LV after 96 h. The upper part shows brightfield images, the lower part, green fluorescence intensity. Downregulation of GFP (sh*GFP*) decreases green fluorescence intensity of cells without affecting cell morphology or proliferation. sh*CALB2*-transduced cells show no changes in green fluorescence intensity. However, a change in cell morphology (more spindloid cells) and a decrease in viability/proliferation are evident in MSTO-CR-sh*CALB2* cells. Scale bar: 200 μm.

**Figure 2 ijms-19-04015-f002:**
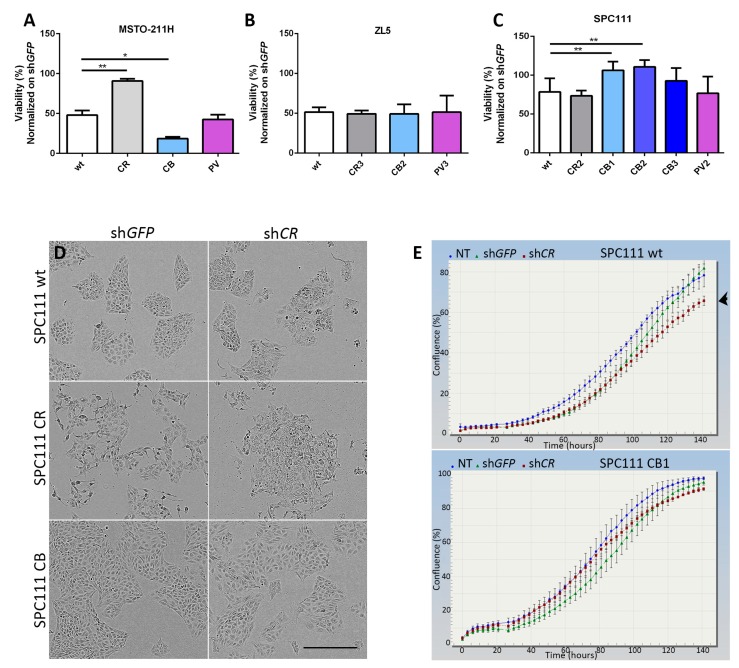
CB rescues the proliferation/viability of SPC111 cells after CR downregulation. Effect of sh*CALB2*-mediated downregulation on cell proliferation/viability of different CaBPs-expressing clones, MSTO-211H (**A**), ZL5 (**B**), and SPC111 (**C**). Signals normalized to sh*GFP*-transduced cells. In SPC111 cells, CB-expressing clones show increased cell viability in comparison to wt cells (n = 4 independent experiments; asterisks represent * *p* ≤ 0.05, ** *p* ≤ 0.01, respectively); scale bar: 400 μm; (**D**) Brightfield images showing the morphology of SPC111 wt, CR- and CB-expressing cells after CR downregulation (left panels: sh*GFP*-treated (control) cells; right panels: sh*CALB2-*treated cells); (**E**) Real-time growth curves (confluence) measured in SPC111 wt and CB-expressing cells after treatment with sh*CALB2* and sh*GFP*, the latter used to normalize the results. Proliferation of SPC111 wt cells was slightly affected by CR downregulation possibly due to a decrease of cell proliferation/viability (arrowhead). CB-expressing cells showed a similar growth curve as the sh*GFP*-treated cells (lower panel). In blue, no treatment (NT); in green, sh*GFP* treatment; in red, sh*CALB2* downregulation.

**Figure 3 ijms-19-04015-f003:**
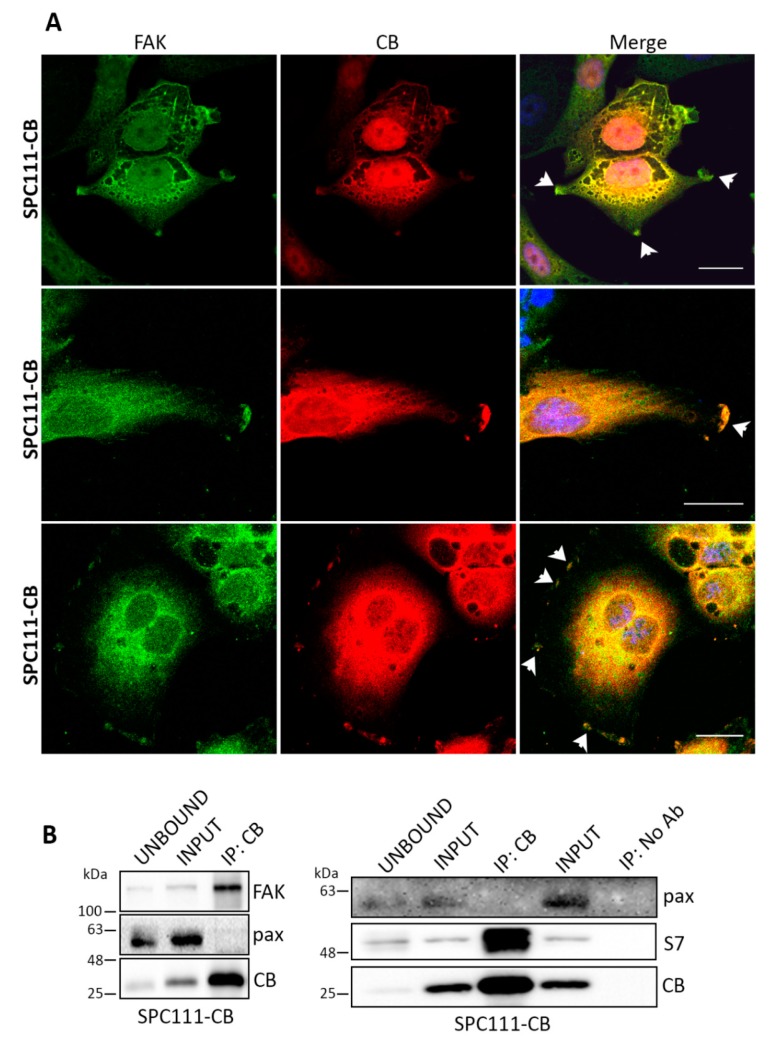
CB co-localizes with focal adhesion kinase (FAK) at focal adhesion sites. Co-IP experiments confirm an interaction of CB with FAK and septin 7. (**A**) Representative confocal images from fixed cells stained for FAK (green) and CB (red) in SPC111-CB cells. CB co-localizes with FAK at the leading edge of the cells and forms dot-like patterns (upper and bottom panels) and membrane ruffles (middle and bottom panels) typical of focal adhesions (see arrowheads) as previously shown for CR [13]. Scale bar: 20 µm; (**B**) Co-IP experiments with cellular lysates from SPC111-CB cells shows co-immunoprecipitation of CB with FAK and septin 7. An input sample collected before immunoprecipitation is shown as INPUT. The INPUT, UNBOUND, as well as the immunoprecipitated (IP) samples were separated using 10% SDS-PAGE and followed by Western blot analysis using anti-CB, anti-FAK, anti-septin 7 and anti-paxillin (negative control) antibodies (*n* = 3 independent experiments). A co-IP with no antibodies is shown as a negative control of the assay.

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
