# Peer review of "Calretinin Functions in Malignant Mesothelioma Cells Cannot Be Replaced by the Closely Related Ca2+-Binding Proteins Calbindin-D28k and Parvalbumin"

_ijms, 2018, doi:10.3390/ijms19124015_

Reviewer 1 Report

The authors present an interesting study where they assessed the capacity of calretinin (CR) to act as both a Ca2+-buffer and seonsor protein. Furthermore, they investigated if other closely related Ca-binding proteins (CaBP), namely clabindin-D28k (CB) and parvalbumin (PV) might substitute for CR function(s) in mesothelioma cells. 

The paper is well presented, easy to read and conveys a logical progression in which appropriate in vitro experiments are performed to assess their hypotheses. Referencing is appropriate.

There are a few minor grammatical/typographical errors listed below that will require correction prior to publication.

Apart from these instances. I am satisfied with the quality of the work and I believe the data presented will be of interest to the mesothelioma research community.

Corrections to consider:

Line 50: symbol missing - ?-tubulin

Line 74: delete 'a' after PV.

Line 323: the use of the word significant indicates a statistical test has been performed, but no stats are shown. Either change the word, or indicated level of statistical significance.

Line 352: Consider changing the word 'proof' to 'prove' or 'confirm'.

Line 355: consider showing endogenous CR levels in sup data, rather than 'not shown'.

Line 423: confusing sentence. Consider punctuation between 'signalling important' in sentence.

Line 427: delete 'also' after CB.

Line 447: revise 'from the ones of'. Consider " in comparison to" or replace with 'from'

Line 452: revise 'to be the presumably". Consider " ...to be the site where CR interacts with its binding partners"

Line 453: space in parentheses after 'second helix'

Author Response

Dear reviewer,

We thank for the helpful comments that has helped to improve the quality of the revised manuscript. We have addressed all the points raised by the reviewer and the answers are listed in a point-to-point response.

Sincerely,

Beat Schwaller

Reviewer #1

Line 50: symbol missing - ?-tubulin

This has been corrected.

Line 74: delete 'a' after PV.           

This has been corrected.

Line 323: the use of the word significant indicates a statistical test has been performed, but no stats are shown. Either change the word, or indicated level of statistical significance.

The word was eliminated.

Line 352: Consider changing the word 'proof' to 'prove' or 'confirm'.

The word was changed.

Line 355: consider showing endogenous CR levels in sup data, rather than 'not shown'.

This has already been shown in a previous study by Blum et al. (2013)  (Ref # 20). This is mentioned in the revised Ms.

Line 423: confusing sentence. Consider punctuation between 'signalling important' in sentence.

We have modified the sentence.

Line 427: delete 'also' after CB.

The sentence was changed.

Line 447: revise 'from the ones of'. Consider " in comparison to" or replace with 'from'

The sentence was modified.

Line 452: revise 'to be the presumably". Consider " ...to be the site where CR interacts with its binding partners"

The sentence was modified.

Line 453: space in parentheses after 'second helix'

This was corrected.

Reviewer 2 Report

Calretinin (CR), Calbindin D28K (CB) and Parvalbumin (PV) belong to the family of EF-hand Ca2+-binding proteins (CaBP) and act as Ca2+-buffering proteins. However, on top of that, CR and CB play a crucial role as intracellular Ca2+ sensors. Considering that CR is a known biomarker of human malignant mesothelioma (MM) and plays a critical role in carcinogenesis and proliferation of MM cells, the authors in this manuscript, investigated its function on cell survival by assessing its functional redundancy through overexpression of the closely related CaBPs, CB and PV in human MM cells. The authors generated stable MM cell lines overexpressing the specific CaBPs while downregulating endogenous CR.

The authors demonstrated that in cells overexpressing high to moderate CR endogenously (MSTO-211H and ZL5 cells), overexpression of CB or PV did not rescue cell death after CR downregulation. However, in SPC111 cells that expressed very low amount of CR, CB but not PV was able to rescue cell proliferation and morphology observed after CR downregulation. These data suggested that CaBPs role as Ca2+ sensors and interaction with key proteins involved in cell proliferation survival such as FAK and Septin-7, shown here, might be more relevant functions to promote MM cell migration or invasion. The authors raised a relevant hypothesis and demonstrated here the unique role of CR in MM cells that cannot be substituted by the closely related CaBPs CB and PV. The data presented here increased the knowledge in the field of MM cell survival and therapy, however the experimental design and data interpretation are often superficial, and the authors need to address the following issues:

1- It is excellent the authors were able to titrate levels of CaBP overexpression in SPC111 cells, I don't understand the rationale in using this cell line since they express the lowest level of  CR. Why not focus on MSTO-211H cells where the knockdown of CR has more biological relevance and overexpress stoichiometric amount of the homologous CaBPs.  

2- The authors should show levels of endogenous CR in SPC111 cells  (WT) (Figure 1) and compare it to the cells stably overexpressing the CaBPs in order to determine the expression levels relative to endogenous amounts observed in MSTO-211H cell for example.

3- There is a discrepancy between Figure 1D legend and the text (line 350) regarding the duration of the shRNA treatment (72h vs. 96h treatment)

4- line 371, the authors need to correct the figure number calling, the statement refers to Figure 2C instead of Figure 2A right

5- Line 373: is an overstatement when talking about a “dominant negative mutant” effect. The authors need to first determine the level of the overexpressed CaBPs to levels similar to that observed endogenously in MSTO-211H or ZL5 cells regarding calcium buffering capacity and stoichiometry and assess the effect of the CR-GFP compared to CR overexpression to determine whether the GFP fusion hinders protein function beforehand.

6- In the Knockdown experiment in Figure 1E, it seems like the overexpression of murine (?) CR-GFP fusion protein did not rescue the loss if the endogenous human CR. The authors need to add a proper control like LV-GFP overexpression to compare with the effect of CR-GFP overexpression. A western blot panel to show the level of CR and CR-GFP under the various experimental conditions.

7- It is known that excessive overexpression of functional proteins can induce detrimental growth defects through resource overload, stoichiometric imbalance, promiscuous interactions, and pathway modulation associated with the degree of overexpression. In Figure 2D-E, determine by WB whether the CR LV-shRNA used here to knockdown the human endogenous protein has any effect on the level of expression of the recombinant murine CR and CB proteins. Better investigate the “dominant negative mutant” effect by comparing various levels (very low to high) of CR and CB overexpression in MSTO211H cells (Figure 2A). Stoichiometric overexpression of CR should normally rescue loss of endogenous protein.

8- In Figure 2D, determine by western blotting the expression level after of CR KD and/or overexpression and assess whether cells are undergoing apoptosis in the various conditions using Cleaved Caspase 3 or cleaved PARP. These data will help answer the question of whether CB/CR overexpression prevented apoptosis or stimulated proliferation. 9- Limit the number of data not shown; relevant data should be put on the supplemental information.

Author Response

Dear reviewer,

We thank for the helpful comments that has helped to improve the quality of the revised manuscript. We have addressed all the points raised by the reviewer and the answers are listed in a point-to-point response.

Sincerely,

Beat Schwaller

Reviewer #2

1- It is excellent the authors were able to titrate levels of CaBP overexpression in SPC111 cells, I don't understand the rationale in using this cell line since they express the lowest level of CR. Why not focus on MSTO-211H cells where the knockdown of CR has more biological relevance and overexpress stoichiometric amount of the homologous CaBPs.

We agree with the reviewer that studies with MSTO-211H single clones would have been of some advantage. Unfortunately, we have not succeeded to generate MSTO-211H clones from single cells. We have tried this already for the second time, as we had tried also in a previous published study (Blum et al. 2013), also without success. While we have managed to obtain single-cell clones from essentially all cell lines we have tried (MM, but also from different cancer types including colon and prostate cancer), it doesn’t work with MSTO-211H cells.

One advantage to work with the low-CR cell lines is exactly the fact that they have low endogenous levels. When looking at CR IHC stainings in MSTO-211H cells, it is obvious that staining varies from almost negative to very strong (e.g. shown in the supplemental Figure for the reviewer) indicating a large variability of CR expression levels in individual MSTO-211H cells that might have posed problems of interpreting data obtained after shRNA-mediated CR downregulation in MSTO-211H cells. Also of note, the expression of the CaBPs (CR, PV, CB) is achieved by a constitutive LV expression system. Thus, one cannot tune CaBP expression levels to “stoichiometric amounts” of CR. In SPC111 and ZL5 we were able to obtain “similar” expression levels of CR and CB (Fig. 1 and Fig. S1). However this fact was beyond the experimenter’s influence.

2- The authors should show levels of endogenous CR in SPC111 cells (WT) (Figure 1) and compare it to the cells stably overexpressing the CaBPs in order to determine the expression levels relative to endogenous amounts observed in MSTO-211H cell for example.

As already reported in the previous study by Blum et al. (2013) Intl. J. Cancer, endogenous CR expression levels in SPC111 cells are very low, i.e. close to the detection limit of our Western blot assays. Rather large amounts of proteins (50 µg) and long exposure times had to be used to obtain a very weak CR signal (s. Fig. 1a (lower panel) in the above mentioned paper). As suggested by the reviewer we have compared CR expression levels of SPC111-CR and ZL5-CR clones with the endogenous expression levels in MSTO-211H cells. In the overexpressing ones, CR levels are clearly higher than in the MSTO-211H wt cells (Fig. S1B). This is mentioned in the revised Ms. and shown in the new Fig. S1).

3- There is a discrepancy between Figure 1D legend and the text (line 350) regarding the duration of the shRNA treatment (72h vs. 96h treatment)

The duration of shRNA treatment was 96 h (4 days) and we have corrected this discrepancy in the revised Ms.

4- line 371, the authors need to correct the figure number calling, the statement refers to Figure 2C instead of Figure 2A right

We thank the reviewer for the careful reading and we have changed the text in the revised Ms.

5- Line 373: is an overstatement when talking about a “dominant negative mutant” effect. The authors need to first determine the level of the overexpressed CaBPs to levels similar to that observed endogenously in MSTO-211H or ZL5 cells regarding calcium buffering capacity and stoichiometry and assess the effect of the CR-GFP compared to CR overexpression to determine whether the GFP fusion hinders protein function beforehand.

We fully agree with the reviewer. Since it never was our intention to go in this direction in this study and since GFP-CR was essentially used as a control to demonstrate that both, the shGFP and the shCALB2 constructs are functioning properly (as demonstrated in Fig. 1E), we have eliminated this data (old Fig. 2A) and also the statement about possibly “dominant negative effects” in the revised manuscript. Instead we have added the data on the overexpression of CR in MSTO-211H cells and show that in the MSTO-CR cells, CR levels after shCALB2 treatment are considerably decreased (shown in the new Fig. S2C), yet CR levels were still similar or higher than in MSTO wt cells (Fig. S2D). This is the reason why cell growth/viability was not significantly impaired in shCALB2-treated MSTO-CR cells (new Fig. 2A).

6- In the Knockdown experiment in Figure 1E, it seems like the overexpression of murine (?) CR-GFP fusion protein did not rescue the loss if the endogenous human CR. The authors need to add a proper control like LV-GFP overexpression to compare with the effect of CR-GFP overexpression. A western blot panel to show the level of CR and CR-GFP under the various experimental conditions.

The cDNA sequence for CR used in all experiments is from human CR (CALB2) not from mouse (Calb2). The only point we wanted to address in Fig. 1E is that with shGFP down-regulating only GFP-CR (not endogenous CR), the green fluorescence of GFP-CR disappears almost completely and that shCALB2 is not powerful enough to completely downregulate both, endogenous CR and overexpressed GFP-CR, since there are still cells showing rather strong green fluorescence. The presence of some green surviving cells might also indicate lower efficiency of infection with the shCALB2 LV in the MSTO GFP-CR cells. In agreement with this hypothesis, the fluorescence intensity of green fluorescence in the surviving cells is almost the same as in untreated MSTO GFP-CR cells. Nonetheless the relevant point is that the shRNA approach is functioning as also shown for CR in MSTO wt cells (Fig. 1D).

Also of importance, endogenous CR levels are not affected by the shGFP as shown in Fig. 1D, while CR is significantly reduced in MSTO-CR cells treated with shCALB2 (Fig. S2C).

7- It is known that excessive overexpression of functional proteins can induce detrimental growth defects through resource overload, stoichiometric imbalance, promiscuous interactions, and pathway modulation associated with the degree of overexpression. In Figure 2D-E, determine by WB whether the CR LV-shRNA used here to knockdown the human endogenous protein has any effect on the level of expression of the recombinant murine CR and CB proteins. Better investigate the “dominant negative mutant” effect by comparing various levels (very low to high) of CR and CB overexpression in MSTO211H cells (Figure 2A). Stoichiometric overexpression of CR should normally rescue loss of endogenous protein.

We fully agree with reviewer on all the above-mentioned mechanisms that may be caused by overexpression of a protein. In particular, if this is a truncated protein or a fusion protein containing a “foreign” protein such as GFP or similar. Thus, such a protein as GFP-CR was essentially used as a control to demonstrate the effectiveness of the shGFP RNA and that this RNA has no detrimental effect per se and is thus useful as a control for putative effects that might be caused by the LV infection of MM cells.

Moreover, we don’t investigate “absolute” effects, i.e. the effect of overexpression of e.g. CB or PV. We mostly focus on the question, whether downregulation of CR in CB- or PV-expressing cells shows the same phenotype as the CR down-regulation in the parental (wt) cells. If not, we assume that the rescue of the CR down-regulation is resulting from the presence of the substitutes CB or PV. Since we cannot control for CaBP expression levels with our constitutive LV expression system, it might be that we “miss” some of the protective effects, because we are not in the optimal range for that particular CaBP in order to “rescue”. However such a stoichiometric comparison is beyond the scope of this short report.

8- In Figure 2D, determine by western blotting the expression level after of CR KD and/or overexpression and assess whether cells are undergoing apoptosis in the various conditions using Cleaved Caspase 3 or cleaved PARP. These data will help answer the question of whether CB/CR overexpression prevented apoptosis or stimulated proliferation.

Endogenous levels of CR in SPC111 wt cells are very low, close to the detection limit of our Western blot analysis as previously shown in the paper by Blum et al. (2013) Intl. J. Cancer in Fig. 1a. Thus, it would be essentially impossible to determine the degree of CR downregulation in shCR-SPC111 cells. However, this low amount of CR appears to be still relevant with respect to proliferation, since growth was slightly affected (reduced) after LV-shCALB2 treatment (Fig. 2E). A non-specific LV effect can be excluded, since treatment of cells with LV-shGFP had no effect on proliferation. Visual inspection of wells recorded with the Incucyte system showed no obvious increase in dead/floating cells in shCALB2-SPC111 wt cells indicating that reducing CR levels by LV-shCALB2 treatment likely results in a very mild impairment of proliferation and possibly a slight increase in apoptosis. This had been previously observed in the sarcomatoid MM cell line ZL34 (Fig. 5e in Blum et al. (2013). The increase in apoptosis in ZL34 cells was however not significant. This is exactly in line with our findings that CR overexpression in CR-negative mouse primary mesothelial cells increases the proliferation rate (Blum et al. (2015) 16:153; Fig. 3). Such an increase is seen in both, mesothelial cells from wt and from CR-/- mice. However, the most important point in Fig. 2E is that CB overexpression abolished (rescued) this mild phenotype, whether this is a proliferation phenotype (most likely) or alternatively a rescue from apoptosis (less likely, but cannot be excluded entirely).

9- Limit the number of data not shown; relevant data should be put on the supplemental information.

We agree with the reviewer and have added supplemental Material as proposed by reviewer #2. This has resulted in new Figs. S1 and S2.

Round  2

Reviewer 2 Report

In this revised version of the manuscript, the authors made significant improvements and answered many of the concerns raised by the reviewers. It is pretty clear that the CaBPs  (CR, CB, and PV) investigated here are not functionally redundant in terms of their effects in the proliferation and survival of the human malignant mesothelioma cells lines used here. I thank the authors for adding the supplemental information needed for a better grasp of the manuscript.